# Approximating Interactive Human Evaluation with Self-Play for Open-Domain Dialog Systems

**Asma Ghandeharioun**,* **Judy Hanwen Shen**\*, **Natasha Jaques**\*,
**Craig Ferguson, Noah Jones, Agata Lapedriza, Rosalind Picard**
Department of Media Arts and Science
Massachusetts Institute of Technology
Cambridge, MA 02139
{asma_gh,judyshen,jaquesn}@mit.edu
{fergusoc,ncjones,agata}@mit.edu, picard@media.mit.edu

## Abstract

Building an open-domain conversational agent is a challenging problem. Current evaluation methods, mostly post-hoc judgments of static conversation, do not capture conversation quality in a realistic interactive context. In this paper, we investigate interactive human evaluation and provide evidence for its necessity; we then introduce a novel, model-agnostic, and dataset-agnostic method to approximate it. In particular, we propose a self-play scenario where the dialog system talks to itself and we calculate a combination of proxies such as sentiment and semantic coherence on the conversation trajectory. We show that this metric is capable of capturing the human-rated quality of a dialog model better than any automated metric known to-date, achieving a significant Pearson correlation ($r > .7, p < .05$). To investigate the strengths of this novel metric and interactive evaluation in comparison to state-of-the-art metrics and human evaluation of static conversations, we perform extended experiments with a set of models, including several that make novel improvements to recent hierarchical dialog generation architectures through sentiment and semantic knowledge distillation on the utterance level. Finally, we open-source the interactive evaluation platform we built and the dataset we collected to allow researchers to efficiently deploy and evaluate dialog models.

## 1  Introduction

The goal of an open-domain conversational agent is to carry out natural social interactions with humans. Current state-of-the-art generative neural networks fail in producing key aspects of good natural conversation, including staying on topic, not being repetitive, and generating emotionally appropriate responses. One of the biggest challenges in training better dialog systems relates to the difficulty of evaluating them. Automatic metrics such as BLEU score relate poorly to human judgment of dialog quality [1], and while embedding-distance based metrics provide an alternative [2], we will show that they also do not correlate well with human evaluation. Without a reliable metric to optimize, training high quality dialog models remains difficult.

Since humans are the ultimate authority on what constitutes a good conversation, many authors rely on human ratings to evaluate their methods [3–5]. The predominant procedure for obtaining human ratings uses *static evaluation*: a context of several sentences, often originating from the dataset that dialog model was trained on, is used as input to generate a response (either a single utterance or multiple consecutive utterances). This particular generated response for this particular context is then provided for a human to evaluate. However, such observational evaluation of a static text sample

---

Table 1: Static evaluation fails to capture a lack of diversity in a dialog model's responses, as well as its inability to track the conversation and respond in emotionally appropriate ways. We argue interactive evaluation is needed to evaluate dialog models, and show that our Emotion+Infersent (EI) models trained on a larger and more diverse corpus, produce better interactive dialog. We present strong evidence that our novel dialog self-play framework combined with psychologically motivated novel automated metrics can accurately estimate quality of a model with respect to its ability to carry out multi-turn open-domain conversations.

| Static HRED - Cornell | Interactive HRED - Cornell | Interactive EI HRED - Reddit | Self-play EI HRED - Reddit |
|---|---|---|---|
| [Context]: you're weird. [HRED]: i'm sorry. | [User]: i'm feeling sad today. [HRED]: no. [User]: yes i am! [HRED]: i'm sorry, i'm sorry. i don't want to know. [User]: you're weird. [HRED]: i'm sorry. | [User] talk to me [EI HRED]: it's my job. [User]: i love you [EI HRED]: thank you for the kind words. [User]: do you love me? [EI HRED]: it's a bit personal, so I sent you a dm. [User]: i'm blushing | [EI HRED] i can't wait to see what i can do. [EI HRED] that's a good idea. [EI HRED] how do you deal with people? [EI HRED] i've been on reddit for a long time. [EI HRED] i am in the same boat. |

is limited in capturing common failure modes of open-domain dialog systems, such as a lack of diversity in the responses, inability to track long-term aspects of the conversation, and inability to maintain a consistent persona [6]. Despite this, static evaluation is commonly used for evaluating these exact qualities [7, 8].

Conversation is inherently a process. In this paper, we argue that multi-turn interactive human evaluation is essential for evaluating this process, and for making progress in improving open-domain dialog models. A multi-turn interactive evaluation is open-ended, and allows the users to derive and control the trajectory of the conversation by saying whatever they want. As opposed to post-hoc observations of a generated response to a static context, this is an ultimate test of generalization. Table 1 illustrates an example where a bot generates a coherent static response, but interactive evaluation shows that the distribution of its responses has collapsed onto repeatedly saying the same phrase, a common problem in open domain dialog [9].

The relative sparsity of interactive human evaluation of dialog systems may relate to the difficulty and expense of collecting human data. Therefore, we develop a way to approximate human judgment of interactive dialog quality using a novel form of dialog self-play. We begin by moving beyond superficial word-level mapping by proposing a series of metrics to evaluate the quality of conversation motivated by findings in psychology. Specifically, inspired by the effectiveness of sense of humor in creating solidarity [10], style matching for forming relationship stability and social cohesiveness [11, 12], and the importance of active listening through forming follow up questions [13], we propose metrics to capture sentiment, semantics, and user engagement. We then fit a function that predicts human assessments of conversation quality given these metrics. This function is used to predict bot quality through self-play: for a fixed number of turns, the bot generates utterances which are fed back into itself as input in the next turn. The same metrics described above are computed on the self-play generated conversation, and the same function fit to human data is used to predict the bot quality. We show a very high Pearson correlation ($r > .7, p < .05$) between the predicted quality scores and the ground-truth human judgments of bot quality, suggesting self-play is a good proxy for interactive conversation assessment.

To demonstrate the relevance of the interactive evaluation and the proposed self-play evaluation, we perform extended experiments with different hierarchical architectures. In particular, we compare three recent hierarchical baselines: HRED [5], VHRED [3], VHCR [4]. Motivated by sentiment and semantics being key aspects of producing high quality conversations, we regularize the top level of the hierarchy to ensure it encodes such information, using model distillation [14]. Our results show the effectiveness of the proposed regularization in interactive evaluation in both the human-bot and the self-play scenarios.

This paper makes three main contributions: 1) demonstrates the necessity of multi-turn interactive evaluation to capture the quality of the dialog systems; 2) presents a novel self-play framework to estimate a new psychology-motivated hybrid quality score. These estimations are highly correlated with quality scores obtained from interactive human evaluation, more strongly than the state-of-the-art automated metrics; 3) proposes a new method of regularizing hierarchical seq2seq models with knowledge distillation. All the code, data, and interactive evaluation platform resulting from our work are publicly available.

## 2 Related work

Interactive evaluation in dialog has been mostly limited to presenting the results of competitions (e.g. the Alexa prize [15, 16], or the Conversational Intelligence Challenge [6]). Those findings reveal that most bots do not perform well in interactive evaluation, due to repetitiveness, inability to balance dialog acts across the conversation, and inability to maintain a consistent persona [6]. Even work aimed at maintaining a persona does not test in an interactive setting [7, 8]. To the best of our knowledge, no prior work has compared multi-turn, interactive human evaluations of open-domain dialog models to traditional forms of evaluation.

Dialog systems remain difficult to train due to the lack of metrics that can effectively capture good dialog quality. Several authors have proposed training automatic predictors of human judgment or to combine human judgment with automatic metrics [17–19]. However, a state-of-the-art model trained to predict human judgments achieved a Pearson correlation of .44 with the ground truth [18].

Perhaps the lack of research into interactive evaluation relates to the difficulty and cost of collecting human ratings. We show that human judgments of the quality of an interactive evaluation can be automatically and reliably approximated using dialog model self-play. There is limited work investigating self-play for dialog systems: Shah et al. [20] use a task schema and user simulator to generate samples for input to a goal-directed dialog system, while Li et al. [9] use a copy of a dialog model to compute a reward function that can be optimized with reinforcement learning. However, we are not aware of prior work using self-play for approximating interactive human evaluation.

Interactive conversation necessitates tracking long-term aspects of the dialog like the topic and tone. Hierarchical recurrent neural networks (RNNs) have been proposed as a way to improve long-term tracking of the conversation, through maintaining both a word- and utterance-level RNN [3–5, 21, 22]. Yet dialog is more than language modeling, it requires topic and social coherence. Prior performance improvements to dialog models using topic information include appending topic as an additional input [23], or extracting topic information using Latent Dirichlet Allocation [24, 25]. Towards social and emotional coherence, previous works have investigated various features and loss functions based on emotion [26–30]. Given research highlighting the ineffectiveness of LDA for short texts [31], such as those involved in casual conversation, and the unavailability of topic and tone supervision at-scale, approaches overcoming these limitations are preferred. To the best of our knowledge, transferring sentiment and semantic information from a pre-trained model directly into a dialog model using knowledge distillation [14] has not been studied. Thus, we select a set of recent hierarchical dialog models and their improved versions through knowledge distillation for a thorough multi-turn interactive evaluation and comparison to traditional evaluation.

## 3 Knowledge distillation for sentiment and semantic regularization

To systematically compare multi-turn interactive evaluation of open-domain dialog with traditional forms of evaluation, we include a diverse set of models. Particularly, we build on three existing hierarchical seq2seq architectures designed for dialog. Here, we provide a brief summary; for detailed information, see [5, 3, 4]. The first baseline model, Hierarchical Recurrent Encoder Decoder (HRED) [5] extends a traditional seq2seq model by adding a third recurrent neural network (RNN), which is only updated after each dialog turn, or utterance. The idea behind this *Context RNN* is that it could potentially track longer term aspects of the conversation, such as the topic; however, there is no guarantee that it will learn to do so. The decoder of the HRED model conditions on both the embedding produced by the encoder for the current utterance, $h_n^e$, and the embedding of the Context RNN for the previous utterance, $h_{n-1}^c$.

The second baseline model, Variational HRED (VHRED) [3], extends HRED with a variational constraint on the utterance embedding space $z$. Let $x_n = [w_{1n}, w_{2n} \ldots w_{mn}]$ be the $n$-th utterance composed of tokens $w_{1..m}$. VHRED predicts $x_n$ as follows:

$$h_n^e = f^e(x_{n-1}) \tag{1}$$

$$h_{n-1}^c = f^c(x_{n-1}, h_{n-1}^e) \tag{2}$$

$$\mu, \Sigma = f(h_{n-1}^c) \tag{3}$$

$$p_\theta(z_n|x_{<n}) = N(z|\mu, \Sigma) \tag{4}$$

$$p(x_n|x_{<n}) = f^d(h_{n-1}^c, z_n) \tag{5}$$

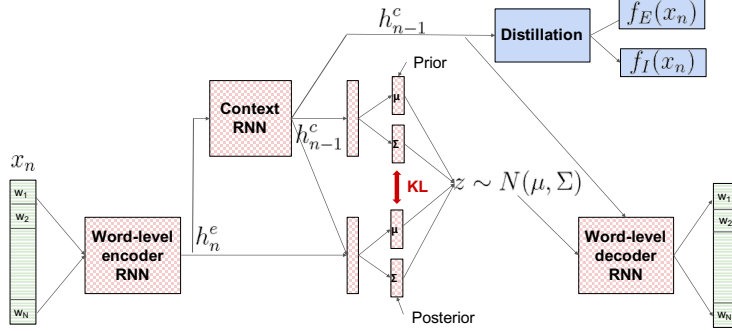

Figure 1: Illustration of the EI regularization (blue-solid) applied to VHRED baseline (red-checkered) to enforce encoding sentiment and semantics of an utterance in the Context RNN. The EI regularization can be similarly applied to HRED and VHCR.

Equations (1)-(5) describe the computation of VHRED at inference time where $f^e$, $f^c$, and $f^d$ are Gated Recurrent Unit (GRU) networks for the encoder, context, and decoder RNNs, respectively; at training time, it allows the computation of $z$, $\mu$, and $\Sigma$ to condition on the encoding of the target utterance, $h_n^e$, giving the posterior distribution $p_\Psi(z_n|x_{\leq n})$. A Kullback-Leibler (KL) divergence constraint is placed between the posterior and prior, $D_{KL}(p_\Psi||p_\theta)$.

The third model, Variational Hierarchical Conversation RNN (VHCR) [4] further extends VHRED by drawing a prior encoding $z^{conv} \sim N(0, I)$ for each conversation, allowing all parts of the model ($f^c, \mu, \Sigma$) to condition on $z^{conv}$, which is unchanging throughout the conversation.

### 3.1 Emotion and Infersent regularization (EI)

While the hierarchical design of these models is motivated by a desire to allow tracking high-level, slow-changing aspects of the conversation like topic or tone, it is unclear that the network will be able to model these aspects without additional structure or information. We thus propose a regularization to the top level of the hierarchy, the Context RNN, to force it to encode both the sentiment and semantics of the utterance. To do this, we leverage a state-of-the-art sentiment detection model trained on a large Twitter corpus [32], as well as the recently proposed *Infersent* sentence-embedding model trained to predict the meaning (i.e. entailment, contradiction) of sentences [33], and distill them into the *Context RNN*.

First, we use these models to predict the emotional content, $f_E(x_n)$, and infersent embedding, $f_I(x_n)$ of each input utterance. We then add an additional network to the hierarchical models which predicts these values based on the context RNN embedding of the utterance: $f^{distill}(h_n^c) = < f_E(x_n), f_I(x_n) >$. The goal is to transfer knowledge of emotion and semantics in text into the context RNN via knowledge distillation [14].

Figure 1 illustrates, in blue color, the EI regularization applied to the VHRED model. The regularization can be similarly applied to HRED and VHCR. In our experiments we refer to the regularized models as HRED-EI, VHRED-EI, and VHCR-EI, respectively, or, more generally, EI models as opposed to baseline models. The code for all our models is available at `https://github.com/natashamjaques/neural_chat` and was originally based on [4]. For details regarding hyper-parameter tuning refer to §A.12.

## 4 Interactive evaluation methodologies

### 4.1 Traditional evaluation

**Automatic metrics** Embedding-based metrics compare generated sentences to ground truth sentences using a vector representation of words [2]. In this work, we use three embedding metrics: embedding *average*, vector *extrema*, and *greedy* matching. These three metrics are used in previous open-domain dialog models [1, 3, 4]. We also use *perplexity* as a standard measure of the likelihood of the

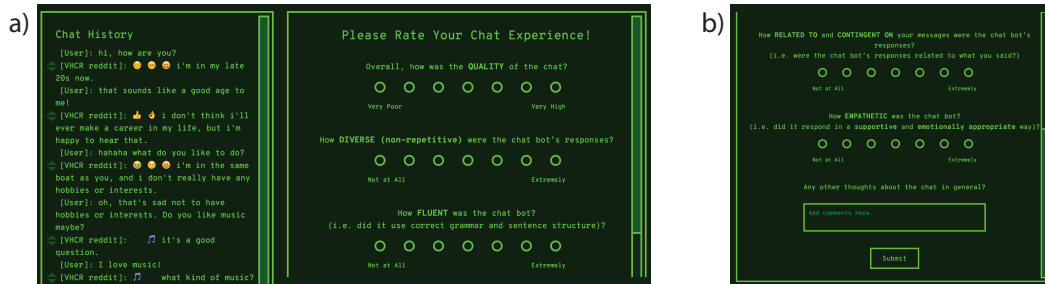

Figure 2: Screenshots of our Interactive Evaluation Platform (available at `https://neural.chat`): (a) chat window (left) and first part of the evaluation form (right); (b) second part of the evaluation form (to show all evaluation questions asked).

generated sentences with respect to the target outputs. Another common metric for variational models is the KL-Divergence between the posterior and the prior distribution, as a way of assessing the information encoded into the latent variables [21] (Figure 1 illustrates KL for the VHRED model). More information regarding embedding metrics can be found in §A.7.

**Conventional static human evaluation** We employ a similar method to previous work for our static human evaluation of generated responses [3, 4], sampling contexts from each corpus and asking humans to compare the generated responses. To reduce ambiguity, we exclude contexts shorter than 10 tokens and contexts containing <unknown> tokens. We recruited participants from Amazon Mechanical Turk (AMT) to compare generated sentences. Annotators could also select a third "tied" option. For each example (context and pair of generated sentences), we asked annotators to compare generated sentences based on quality, fluency, diversity, contingency, and empathy. Each batch of 100 pairwise comparisons were labeled by 6 - 8 annotators.

## 4.2 Interactive human evaluation

To address the limitations of static human evaluation, we built a platform for conducting interactive evaluation of dialog models with humans, which we make available in open-source to the community (see Figure 2). Annotators rated quality, fluency, diversity, relatedness, and empathy of a bot after interacting with it for at least 3 turns. Participants can also upvote or downvote each bot response. For more information about this platform, see §A.10. Our goal is to make this work transparent and reproducible, while adding diversity to the platforms future practitioners can choose to use (e.g. ParlAI [34], Plato Research Dialog System [35], ChatEval [36]).

## 4.3 Novel metrics and self-play

Inspired by real-world human interactions, we introduce novel metrics to capture the morphology of a conversation, i.e., how the users' responses progress over time and how the bot's responses interact with them. We propose a hybrid combination of these metrics, $M_H$, that is optimized to predict conversation quality on human data. We then apply $M_H$ to self-play, i.e., the trajectory of bot-generated responses, and investigate how it relates to human ratings of conversation quality.

**Sentiment metrics** To approximate emotional tone of an utterance, we use a state-of-the-art sentiment detector trained on a large Twitter corpus [32]. This pre-trained model outputs an emotion embedding – a probability distribution over 64 most-frequently used emojis. To estimate the *Sentiment Coherence* between user's query and generated samples, we calculate the cosine similarity between their emotion embeddings. We define a set of weights over the 64 emojis and calculate the weighted sum over an emotion embedding vector to derive a *Sentiment* score which is higher for positive sentiment and lower for negative sentiment (See §A.11). We define *Sentiment Transition* as the change between user's *Sentiment* before and after a bot response. Additionally, *Sentiment Min-Max* is defined by the slope of change between min and max *Sentiment* in user utterances over the course of a conversation. Since humor can be used to create solidarity [10], we count the number of "ha"s in the user response as a proxy for *Laughter*. The combination of these metrics provides a snapshot of the trajectory of sentiment in a conversation and quantifies if the bot is able to elicit positive emotions in the user.

**Semantic metrics** Language style matching is a strong predictor of relationship stability [11] and social cohesiveness [12]; thus, we introduce metrics to capture lexical similarity. We use *Infersent*, a state-of-the-art sentence-embedding model to encode the user and bot responses into a 4096-dimensional embedding space [33]. Infersent was trained to distinguish if two sentences are supporting, contradicting, or have a neutral relationship. We estimate *Semantic Similarity* by calculating the cosine similarity between the infersent embedding of the user's query and the generated bot sample. Additionally, we use the classic Word2Vec embeddings trained on Google News Corpus along with average, extrema, and greedy aggregation methods similar to Section 4.1 to derive *Average Word Coherence*, *Extrema Word Coherence*, and *Greedy Word Coherence* between user and bot responses.

**Engagement metrics** Asking questions is an important active listening skill which is linked to conversation management, attentiveness, and responsiveness [13, 37]. Therefore, we define *Question Score* to quantify if the bot is using question words and/or a question mark. We also introduce *# Words* as a proxy for user engagement that counts the number of words in their response.

**Hybrid metric** ($M_H$) We combine the aforementioned metrics ($M_i$) using linear regression, and optimize their coefficients ($\lambda_i$) to best predict human judgment of interactive conversation quality: $M_H = \sum \lambda_i * M_i + \lambda_0$. We use a leave-one-bot-out scenario where we isolate all the human conversations with one of the dialog models, $\chi_j$, as the hold-out test set. We train the $\lambda_{i,j}$ on the remaining quality ratings. We found that the learned $\lambda_i$s were stable across the training folds, only exhibiting small variations. Other researchers are encouraged to use our learned coefficients directly or adjust them according to their own interactive human evaluation dataset. See §A.2 for more details about the learned $\lambda_i$s.

**Self-play as an approximation for interactive evaluation** Since interactive human evaluation is costly, we propose a *self-play* scenario where the dialog system talks to itself, i.e. the bot generated responses are fed back into it as the next turn input. For each model $\chi_j$, we generate 100 random conversations, fixed at 10 turns. The self-play trajectories created using model $\chi_j$ are treated as the hold-out set. Therefore, the trained $\lambda_{i,j}$ values based on all conversations except for the ones with $\chi_j$ are used to calculate $M_H$ on each generated bot-bot conversation trajectory for $\chi_j$. The estimated $M_H$ values are averaged across conversation samples for $\chi_j$. This value is used for comparison against the ground-truth interactive quality ratings aggregated on the bot-level.

# 5 Experiments

## 5.1 Datasets

A common source of data for open-domain dialog systems is movie scripts, among which the CORNELL dataset [38] is the largest and most commonly used. Therefore, we use it to benchmark against previous state-of-the-art results [4]. Its median conversation length is 3 utterances and the conversations are strictly between pairs of speakers. Recognizing that movie lines have limited conversation diversity, we also built a new corpus, REDDIT. Between the many different subreddits available, the conversations vastly differ on topic, language style, and participation patterns. We select the Casual Conversations forum (`r/CasualConversations`), a community of $607K$ conversationalists discussing a variety of topics. We collect a dataset of $109K$ conversations of at least 3 turns with the median conversation containing 7 utterances from conversational exchanges on the platform in 2018[2]. More more details about this dataset refer to §A.6.

## 5.2 Interactive human evaluation

Table 1 (in §1) illustrates how EI regularization produces a higher quality conversation when compared to baseline. Rather than cherry-picking results, we make all of the bots evaluated in the study available at `https://neural.chat/BRFZACDCOA/` for readers to assess interactively.

Table 2 summarizes human ratings of baseline and EI models obtained via interactive evaluation. In total, 565 ratings were captured. Each dialog model has been evaluated by a number of annotators, ranging from 36 to 56. For additional information about human annotators refer to §A.9. We ran a 3-factor ANOVA on the sum of user scores, where the independent variables are model architecture (HRED, VHRED, VHCR), EI regularization (Baseline, EI), and dataset (CORNELL,

Table 2: Mean human ratings for Baseline and EI (Emotion+Infersent) models for HRED, VHRED, and VHCR architectures with 90% confidence intervals. See §5.2 for 3-factor ANOVA results.

| Model | Metric | Cornell | | Reddit | |
|---|---|---|---|---|---|
| | | Baseline | EI | Baseline | EI |
| HRED | quality | $2.182 \pm 0.305$ | $\mathbf{2.347} \pm 0.313$ | $2.527 \pm 0.310$ | $\mathbf{2.714} \pm 0.299$ |
| | fluency | $3.909 \pm 0.387$ | $\mathbf{4.000} \pm 0.381$ | $4.436 \pm 0.349$ | $\mathbf{4.786} \pm 0.316$ |
| | diversity | $\mathbf{2.836} \pm 0.374$ | $2.735 \pm 0.380$ | $3.418 \pm 0.386$ | $\mathbf{3.554} \pm 0.372$ |
| | contingency | $2.200 \pm 0.291$ | $\mathbf{2.469} \pm 0.336$ | $2.382 \pm 0.288$ | $\mathbf{2.536} \pm 0.322$ |
| | empathy | $\mathbf{2.673} \pm 0.352$ | $2.490 \pm 0.350$ | $3.018 \pm 0.329$ | $\mathbf{3.107} \pm 0.337$ |
| VHRED | quality | $2.022 \pm 0.309$ | $\mathbf{2.333} \pm 0.252$ | $2.694 \pm 0.392$ | $\mathbf{2.864} \pm 0.341$ |
| | fluency | $3.109 \pm 0.351$ | $\mathbf{3.949} \pm 0.396$ | $4.250 \pm 0.496$ | $\mathbf{4.477} \pm 0.402$ |
| | diversity | $3.565 \pm 0.442$ | $\mathbf{4.385} \pm 0.371$ | $\mathbf{5.00} \pm 0.468$ | $4.705 \pm 0.353$ |
| | contingency | $2.261 \pm 0.287$ | $\mathbf{2.487} \pm 0.346$ | $2.472 \pm 0.362$ | $\mathbf{2.773} \pm 0.370$ |
| | empathy | $\mathbf{2.739} \pm 0.374$ | $2.564 \pm 0.367$ | $3.000 \pm 0.393$ | $\mathbf{3.341} \pm 0.385$ |
| VHCR | quality | $2.132 \pm 0.247$ | $\mathbf{2.548} \pm 0.380$ | $2.615 \pm 0.350$ | $\mathbf{2.692} \pm 0.298$ |
| | fluency | $2.679 \pm 0.306$ | $\mathbf{3.976} \pm 0.380$ | $3.923 \pm 0.433$ | $\mathbf{4.308} \pm 0.395$ |
| | diversity | $3.755 \pm 0.340$ | $\mathbf{4.238} \pm 0.421$ | $\mathbf{4.436} \pm 0.455$ | $4.231 \pm 0.382$ |
| | contingency | $2.189 \pm 0.270$ | $\mathbf{2.571} \pm 0.356$ | $2.077 \pm 0.298$ | $\mathbf{2.692} \pm 0.354$ |
| | empathy | $2.340 \pm 0.316$ | $\mathbf{2.714} \pm 0.368$ | $2.974 \pm 0.434$ | $\mathbf{3.288} \pm 0.379$ |

Table 3: Results of automatic traditional metrics for 1-turn responses of models per context of baseline and EI (Emotion + Infersent) models. PPL: perplexity, KL: KL divergence, Avg: Average, Ext: Extrema, Grd: Greedy

| Model | Version | Cornell | | | | | Reddit | | | | |
|---|---|---|---|---|---|---|---|---|---|---|---|
| | | PPL | KL | Avg | Ext | Grd | PPL | KL | Avg | Ext | Grd |
| HRED | baseline | 52.311 | - | .471 | .329 | .331 | 41.730 | - | .649 | .394 | .474 |
| | EI | **47.636** | - | **.560** | **.383** | **.400** | **41.245** | - | **.651** | **.398** | **.482** |
| VHRED | baseline | **49.414** | .264 | .539 | .352 | **.395** | 36.240 | **.188** | .635 | .383 | .464 |
| | EI | 50.526 | **.517** | **.545** | **.355** | .394 | **35.510** | .167 | **.636** | **.392** | **.465** |
| VHCR | baseline | 61.000 | **.562** | .532 | .345 | .382 | **36.736** | **.267** | .619 | .371 | .448 |
| | EI | **49.243** | .475 | **.588** | **.369** | **.444** | 37.198 | .231 | **.639** | **.394** | **.469** |

REDDIT). We found a significant main effect of EI regularization and dataset, but no significant difference between the three types of hierarchical models. We found that adding emotion and infersent (EI) regularization to baseline models improved the interactive chat experience significantly, $F(554, 1) = 9.016, p = .003$. Further, the models trained on the REDDIT dataset performed significantly better, $F(554, 1) = 30.796, p < .001$. This finding validates the hypothesis that distilling information about topic and tone into the top level of the hierarchy is useful for good conversation, and suggests that the REDDIT dataset could provide more realistic training for open-domain dialog and be valuable to the community. Additional ablation results are provided in §A.1.

## 5.3 Traditional metrics

**Automatic metrics** Several prior works have focused on ensuring that the variational KL term remains high in order to improve model quality (e.g. [4, 21]). However, we observe there is no consistency between human quality rating and KL (Table 3). See §A.8 for details about other human metrics, e.g. fluency, diversity, contingency, and empathy. Thus, it is not evident that KL captures human judgements of dialog quality. Even perplexity (a transformation of the cross-entropy loss used to train our models) falls short of capturing human quality judgments, underscoring the difficulty in effectively training good language models. We find embedding metrics show more promise in preserving the order of human quality ratings, but have only weak correlation with human ratings. We present evidence for our novel hybrid metric being a much stronger alternative.

**Human static evaluation** As shown in Table 4, while static human evaluation suggests EI regularization is effective due to a higher number of win judgments[3], the results are noisy and difficult to interpret due to large confidence intervals and a high percentage of ties. The median inter-annotator agreement measured pairwise through Cohen's $\kappa$ [39] for our human evaluation was only 0.176 and 0.120 for CORNELL and REDDIT respectively. This level of annotator agreement is lower than the

Table 4: Results from human static evaluation for EI (Emotion+Infersent) vs. BL (baseline) models as measured by pairwise comparisons of **Quality** with 90% confidence intervals.

| | Cornell | | | Reddit | | |
|---|---|---|---|---|---|---|
| Model | Wins % | Losses % | Ties % | Wins % | Losses % | Ties % |
| HRED-EI | **40.8** $\pm$ 4.9 | 24.5 $\pm$ 4.9 | 34.8 $\pm$ 9.2 | **31.3** $\pm$ 5.2 | 29.5 $\pm$ 6.6 | 39.3 $\pm$ 10.7 |
| VHRED-EI | **36.9** $\pm$ 4.7 | 36.6 $\pm$ 5.6 | 26.6 $\pm$ 6.9 | **39.0** $\pm$ 7.0 | 34.0 $\pm$ 5.3 | 27.0 $\pm$ 8.9 |
| VHCR-EI | **33.0** $\pm$ 6.1 | 29.0 $\pm$ 5.4 | 38.0 $\pm$ 10.1 | **33.7** $\pm$ 7.9 | 27.3 $\pm$ 3.3 | 39.0 $\pm$ 8.6 |

median Cohen's $\kappa$ of previous work [1] and explains the larger confidence intervals. Even after removing ambiguous examples (i.e. where equal number of annotators select each response as being better), large annotation variation persists. This may be due to subjectivity and ambiguity arising from different interpretations of <unknown> tokens or the short length of contexts in the CORNELL corpus (e.g. median length of conversation of 3 utterances). These findings further highlight the importance of an interactive evaluation as opposed to limited static responses.

### 5.4 Novel metrics applied to human data and self-play

We examine how the novel psychologically-inspired metrics relate to the trajectories of the 100 best and 100 worst quality conversations. This is only feasible with interactive evaluation. As shown in Figure 3, we observe that appropriate sentiment, coherent semantics, and engaging users are indispensable to attaining high quality ratings in interactive interaction. Comparing EI and baseline conditions, we see a replication of these trends (Figure 4). For example, EI elicits longer responses from users (greater engagement), with more laughter and higher semantic coherence.

Figure 5 summarizes the relationships between interactive human ratings and the automated metrics[4]. We observe that our sentiment metric applied to human data on its own has higher correlation with interactive human ratings than the commonly used metrics such as perplexity and embedding distance metrics. Most importantly, our novel hybrid metric, $M_H$, applied to self-play [5] aggregated on the model-level is strongly correlated with all human ratings ($r > .7$), while previous metrics achieved $r < .5$. This is a significant finding, suggesting that even without running interactive human evaluation, we can automatically approximate it through self-play. This metric is agnostic to the training set and model type and can be calculated on the trajectory of self-play utterances for any chatbot, regardless of its architecture. One interpretation is that the self-play framework keeps the conversation within the training set distribution, and the model is less likely to produce <unknown> tokens. Therefore, $M_H$ and its sub-components have meaningful values and can be useful for quality approximation.

On a realistic conversation trajectory, $M_H$ is a hybrid of conflicting objectives and thus is less susceptible to exploitation [40]. However, the purpose of the self-play metric ($\hat{M}_H$) in its current form is a post-hoc evaluation of a dialog model. There are precautions if one intends to directly optimize for $\hat{M}_H$ or its sub-components, for example in a reinforcement learning scenario. The current formulation of self-play uses trajectories entirely generated by the same model. If one intends to optimize $\hat{M}_H$, we suggest calculating it on conversation trajectories between the bot and an external baseline model or a fixed copy [41], or adopting adversarial learning by maintaining a discriminator to distinguish between real/fake conversations [42]. This implicitly enforces generating realistic language. Additionally, we have shown how to successfully learn using sub-components of $\hat{M}_H$ as reward functions [43].

## 6 Conclusions

A major obstacle in open-domain dialog generation is the predominant optimization of an objective function that does not closely match human judgment of conversation quality in a naturalistic chat. In this paper, we have argued that it is necessary to go beyond static evaluation by investigating the

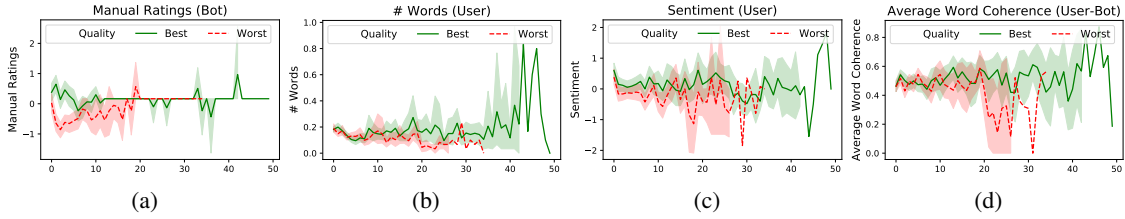

(a)      (b)      (c)      (d)

Figure 3: One hundred highest vs. lowest quality conversation trajectories; lines: mean, shaded area: 90% confidence intervals, x-axis: conversation turns. (a) Timing of upvote/downvote ratings: A bad first impression impedes overall rating. (b) Participants talk longer and use more words in conversations rated higher. (c) High-quality conversations elicit more positive user sentiment; many participants leave after expressing negative sentiment. (d) High-quality conversations are more semantically similar as measured by average word coherence between user query and bot responses. Users tend to leave the conversation when the bot responses are semantically dissimilar.

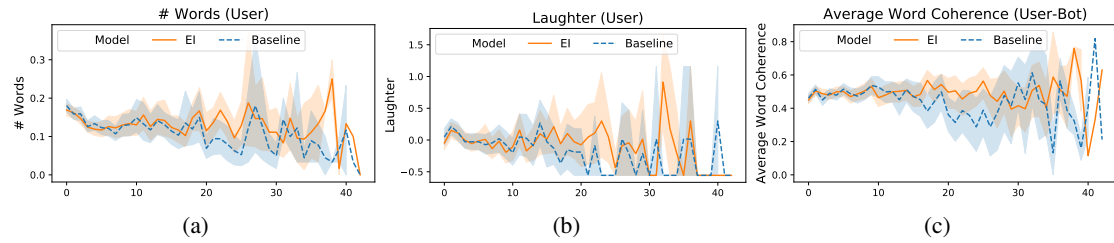

(a)      (b)      (c)

Figure 4: EI vs. baseline conversation trajectories; lines: mean, shaded area: 90% confidence intervals, x-axis: conversation turns. (a) EI elicits longer responses from users, suggesting that they are more engaged compared to the baseline models. (b) EI evokes more laughter from users compared to baseline. (c) EI has higher semantic coherence as measured by average word coherence. The same pattern applies to greedy and extrema word coherence.

| | Automatic M. | | | | | Sentiment M. | | | | | Semantic M. | | | | Engagement M. | | |
|---|---|---|---|---|---|---|---|---|---|---|---|---|---|---|---|---|---|
| | Bits per Word | Perplexity | Average | Extrema | Greedy | Sentiment-U | Sent. Transition-U | Sent. Min-Max-U | Laughter-U | Sent. Coher.-U/B | Semantic Coher.-U/B | Avg. Word Coher. U/B | Extrema Word Coher.-U/B | Greedy Word Coher. -U/B | Question Score-B | # Words-U | Hybrid Metric (M_H) -B/B |
| Quality | -0.145 | -0.145 | 0.141 | 0.100 | 0.132 | 0.214 | -0.089 | -0.038 | 0.044 | 0.069 | -0.010 | 0.035 | -0.007 | 0.042 | 0.013 | 0.048 | 0.725 |
| Diversity | -0.167 | -0.157 | 0.166 | 0.078 | 0.171 | 0.145 | -0.050 | -0.017 | -0.068 | 0.053 | -0.005 | 0.024 | -0.052 | 0.056 | 0.099 | 0.000 | 0.640 |
| Fluency | -0.204 | -0.212 | 0.188 | 0.167 | 0.180 | 0.173 | 0.023 | -0.050 | 0.017 | 0.022 | 0.065 | 0.127 | 0.063 | 0.076 | -0.007 | 0.070 | 0.455 |
| Contingency | -0.046 | -0.047 | 0.059 | 0.079 | 0.060 | 0.177 | -0.083 | -0.051 | 0.034 | 0.034 | -0.044 | 0.033 | 0.030 | 0.029 | 0.005 | 0.039 | 0.263 |
| Empathy | -0.178 | -0.177 | 0.152 | 0.116 | 0.144 | 0.264 | -0.093 | -0.060 | -0.055 | 0.103 | -0.079 | 0.015 | 0.040 | 0.067 | 0.032 | -0.070 | 0.834 |

Figure 5: Pearson correlations between five human metrics and automated metrics. **Sentiment -U** has higher correlation with interactive human ratings than prior metrics. **Hybrid Metric M_H -B/B**, our novel self-play based metric, has higher correlation across all human metrics more than any other metric proposed to-date. **Notes:** -U: Calculated on user response, -B: Calculated on bot response, -U/B: Calculated between user and bot response, -B/B: Calculated between consecutive bot utterances.

strengths of interactive evaluation and highlighting blind-spots of traditional static evaluation methods. To alleviate this problem, we have combined interactive human data with psychologically-motivated measures and introduced a novel hybrid metric. Using this metric in a self-play framework provides results that are strongly correlated with human judgment of chatbot empathy ($r > .8$) and quality ($r > .7$). Additionally, we have demonstrated a significant improvement to several hierarchical seq2seq generative models using regularization of the utterance level of the hierarchy with knowledge distillation. Finally, we have open-sourced the platform together with a new REDDIT dataset.

**Acknowledgments**

We thank Ardavan Saeedi, Max Kleiman-Weiner, Oliver Saunders Wilder, Kyle Kastner, Sebastian Zepf, Ryan Lowe, Abdul Saleh, and Kristy Johnson for helpful discussions, and many others for helping test-drive our bots. We thank the MIT Quest for Intelligence, and MIT Stephen A. Schwarzman College of Computing, Machine Learning Across Disciplines Challenge for providing computing resources, and MIT Media Lab Consortium and RTI2018-095232-B-C22 grant from the Spanish Ministry of Science for supporting this research.

## Footnotes

[2]This REDDIT dataset is available at `https://affect.media.mit.edu/neural_chat/datasets`.

[3]We follow [4] to highlight the higher value between wins/losses and reporting 90% confidence intervals.

[4] For additional correlation results across the human metrics, between $M_i$s and human metrics on a bot-level, and Spearman and Kendall rank coefficients, see §A.3, §A.4, and §A.5 respectively.

[5] Analyzing utterance overlap shows that these self-play conversations are distinct from the training corpus and exhibit high diversity for variational models. Details can be found in §A.13.

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
