[Supplementary Material · supplementary_Dialog_NeurIPS_2019.pdf]

# Approximating Interactive Human Evaluation with Self-Play for Open-Domain Dialog Systems

**Asma Ghandeharioun**,* **Judy Hanwen Shen**∗, **Natasha Jaques**∗,
**Craig Ferguson, Noah Jones, Agata Lapedriza, Rosalind Picard**
Department of Media Arts and Science
Massachusetts Institute of Technology
Cambridge, MA 02139
{asma_gh,judyshen,jaquesn}@mit.edu
{fergusoc,ncjones,agata}@mit.edu, picard@media.mit.edu

## A    Supplementary Materials

### A.1    Ablation models results

We conducted additional evaluations of ablations of our EI models to determine whether emotion or infersent regularization provided the most benefit. The results in Table A.1 reveal that this depends on the dataset and the model in question. We also checked whether simply appending the emotion and infersent embedding of an utterance to the top level of the hierarchy could provide the same benefit as knowledge distillation, even though this would require retaining copies of the DeepMoji and Infersent models, and would be more computationally expensive at inference time. Table A.1 reveals that the *input-only* models do not out-perform the knowledge-distillation EI models on automatic metrics.

Table A.1: Automatic metrics computed on ablations of the EI models, trained with distillation from only the emotion recognition model ($EI_{emo}$), the infersent model ($EI_{inf}$), or receiving emotion and infersent only as input, without knowledge distillation (*input-only*). Whether emotion or semantics provides the most benefit depends on the dataset and the model.

| Model | Version | Cornell | | | | | Reddit | | | | |
|---|---|---|---|---|---|---|---|---|---|---|---|
| | | PPL | KL | Avg | Ext | Grd | PPL | KL | Avg | Ext | Grd |
| HRED | baseline | 52.311 | - | .471 | .329 | .331 | 41.730 | - | .649 | .394 | .474 |
| | input only | 47.911 | - | .549 | .381 | .392 | 41.227 | - | .644 | .395 | .469 |
| | $EI_{emo}$ | 48.619 | - | **.562** | .359 | **.416** | 47.395 | - | .541 | .310 | .371 |
| | $EI_{inf}$ | 47.988 | - | **.562** | .381 | .405 | **41.083** | - | .646 | .394 | .472 |
| | EI | **47.636** | - | .560 | **.383** | .400 | 41.245 | - | **.651** | **.398** | **.482** |
| VHRED | baseline | **49.414** | .264 | .539 | .352 | .395 | 36.240 | .188 | .635 | .383 | .464 |
| | input only | 49.819 | .442 | .543 | .353 | .393 | 40.248 | .312 | .630 | .377 | .456 |
| | $EI_{emo}$ | 51.346 | .636 | **.552** | **.358** | **.401** | 36.212 | .199 | .631 | .380 | .458 |
| | $EI_{inf}$ | 52.143 | **.702** | .539 | .346 | .392 | 36.518 | **.222** | **.637** | .381 | .463 |
| | EI | 50.526 | .517 | .545 | .355 | .394 | **35.510** | .167 | .636 | **.392** | **.465** |
| VHCR | baseline | 61.000 | .562 | .532 | .345 | .382 | **36.736** | .267 | .619 | .371 | .448 |
| | input only | 50.966 | .558 | .531 | .344 | .382 | 37.342 | **.287** | .608 | .365 | .431 |
| | $EI_{emo}$ | 52.407 | .590 | .585 | **.374** | .442 | 37.449 | .254 | .619 | .366 | .444 |
| | $EI_{inf}$ | 53.085 | **.575** | .544 | .356 | .390 | 37.109 | .199 | .629 | .378 | .457 |
| | EI | **49.243** | .475 | **.588** | .369 | **.444** | 37.198 | .231 | **.639** | **.394** | **.469** |

Figure A.1: The learned coefficients ($\lambda_i$) that the hybrid metric ($M_H$) is comprised of. Using a leave-bot-out method, we observe that the $\lambda_i$s are stable. The error bars show 90% confidence intervals.

Figure A.2: Correlation matrix showing the relationships between different aspects of interactive human evaluation. We observe a strong correlation across these aspects.

## A.2 Hybrid metric coefficients

We optimized the coefficients of sub-components of the hybrid metric using a leave-bot-out scenario. As shown in Figure A.1, we observe that $\lambda_i$s are stable across these training iterations. However, because we have optimized a linear regression equation and some of the features have overlapping information, such as different aggregation methods for calculating word coherence, we do not suggest using $\lambda_i$s for direct interpretation; further investigation is required.

## A.3 Human interactive ratings correlation table

Figure A.2 provides detailed information about different metrics from interactive human ratings. We observe that quality is highly correlated with other aspects of the conversation. Specifically, it is most strongly correlated with contingency, which further highlights the importance of semantic metrics of bot-generated responses in a good quality conversation. It also has high correlation with empathy that could better be captured by sentiment metrics.

## A.4 Self-play correlation table

Figure A.3 provides detailed information about the introduced metrics applied to self-play. We observe that several sentiment, semantic, and engagement metrics also transfer to self-play trajectories and the introduced hybrid metric, $M_H$, is highly correlated with human quality ratings aggregated on a bot-level. However, exploiting sentiment or semantic similarity in a self-play scenario should be avoided as it hurts ratings of the model, especially diversity of responses.

## A.5 Additional correlation statistics

Figure A.4 and A.5 provide Spearman's $\rho$ and Kendall's $\tau$ correlation coefficients between human metrics and automated metrics. These tests do not assume a linear correlation as opposed to the Pearson correlation. Similarly to the Pearson correlation results provided in Figure 5, these values provide additional evidence, further confirming the superiority of sentiment metric as well as the newly proposed self-play approximation of the hybrid metric $M_H$.

## A.6 Reddit casual conversation corpus details

Using the 1.7 Billion post comments dataset hosted on Google BigQuery, we extracted post ids for all posts on r/CasualConversation from July 2018 to December 2018. For each post, we built a conversation tree of comments and subsequent replies to extract three-turn dialog. We removed links,

---

[*]Equal contribution

|  | Sentiment -B | Sentiment Transition -B | Sentiment Min-Max. -B | Laughter -B | Sentiment Coher. -B/B | Semantic Coher. -B/B | Average Word Coher. -B/B | Extrema Word Coher. -B/B | Greedy Word Coher. -B/B | Question Score -B | # Words -B | Hybrid Metric ($M_H$) -B/B |
|---|---|---|---|---|---|---|---|---|---|---|---|---|
| Quality -I | 0.716 | 0.258 | -0.231 | 0.811 | -0.155 | 0.092 | 0.695 | -0.364 | 0.055 | -0.739 | 0.751 | 0.725 |
| Diversity -I | 0.626 | 0.159 | -0.161 | 0.440 | -0.787 | -0.594 | 0.360 | -0.861 | -0.636 | -0.536 | 0.792 | 0.640 |
| Fluency -I | 0.536 | 0.144 | -0.284 | 0.676 | 0.187 | 0.354 | 0.448 | 0.013 | 0.242 | -0.508 | 0.354 | 0.455 |
| Contingency -I | 0.311 | 0.360 | 0.159 | 0.387 | -0.027 | -0.112 | 0.319 | -0.056 | 0.097 | -0.374 | 0.281 | 0.263 |
| Empathy -I | 0.852 | 0.059 | -0.502 | 0.862 | -0.157 | 0.008 | 0.503 | -0.484 | -0.115 | -0.813 | 0.724 | 0.834 |

Figure A.3: Correlation matrix showing the relationships between different automated metrics on self-play trajectories and interactive human ratings aggregated on the bot-level. We observe that inducing positive sentiment as measured by Sentiment and Laughter, and being able to generate longer sentences in self-play are associated with higher quality model ratings. It is worth mentioning that maintaining extreme similarity in sentiment or semantics or just asking questions in self-play conversation trajectories could backfire by reducing the diversity of generated responses, though applicable to interactive human data. Most importantly, our novel hybrid metric applied to self-play ($M_H$ -B/B) is highly correlated with all human ratings of the dialog model. **Postfixes:** -I: Interactive human evaluation, -B: Calculated on bot response, -B/B: Metric applied to self-play on two consecutive bot generated utterances when the bot converses with itself.

|  | Bits per word -S | Perplexity -S | Average -S | Extrema -S | Greedy -S | Sentiment -U | Sentiment Transition -U | Sentiment Min-Max -U | Laughter -U | Sentiment Coher. -U/B | Semantic Coher. -U/B | Average Word Coher. -U/B | Extrema Word Coher. -U/B | Greedy Word Coher. -U/B | Question Score -B | # Word -U | Hybrid Metric ($M_H$) -B/B |
|---|---|---|---|---|---|---|---|---|---|---|---|---|---|---|---|---|---|
| Quality -I | -0.151 | -0.151 | 0.146 | 0.117 | 0.129 | 0.219 | -0.066 | -0.043 | 0.023 | 0.067 | 0.019 | 0.036 | 0.003 | 0.050 | 0.022 | 0.041 | 0.678 |
| Diversity -I | -0.160 | -0.160 | 0.109 | 0.047 | 0.104 | 0.153 | -0.076 | -0.025 | -0.080 | 0.039 | 0.005 | 0.030 | -0.047 | 0.057 | 0.124 | -0.002 | 0.636 |
| Fluency -I | -0.183 | -0.183 | 0.218 | 0.193 | 0.191 | 0.168 | 0.006 | -0.051 | -0.003 | 0.007 | 0.092 | 0.110 | 0.056 | 0.055 | -0.021 | 0.092 | 0.420 |
| Contingency -I | -0.040 | -0.040 | 0.063 | 0.075 | 0.058 | 0.164 | -0.095 | -0.076 | 0.009 | 0.042 | -0.007 | 0.036 | 0.054 | 0.041 | 0.030 | 0.015 | 0.161 |
| Empathy -I | -0.170 | -0.170 | 0.160 | 0.126 | 0.155 | 0.277 | -0.103 | -0.078 | -0.047 | 0.104 | -0.069 | 0.016 | 0.047 | 0.073 | 0.029 | -0.058 | 0.755 |

Figure A.4: Spearman correlations between five human metrics and automated metrics. **Sentiment -U** has higher correlation with interactive human ratings than prior metrics. **Hybrid Metric $M_H$ -B/B**, our novel self-play based metric, has higher correlation across all human metrics more than any other metric proposed to-date. **Notes:** -U: Calculated on user response, -B: Calculated on bot response, -U/B: Calculated between user and bot response, -B/B: Calculated between consecutive bot utterances.

excluded [removed] and [deleted] tag comments, and only used text before "*edit*" comments to preserve the original content in the conversation. We make this dataset available for public use at `https://affect.media.mit.edu/neural_chat/datasets`.

Figure A.5: Kendall correlations between five human metrics and automated metrics. **Sentiment -U** has higher correlation with interactive human ratings than prior metrics. **Hybrid Metric** $M_H$ **-B/B**, our novel self-play based metric, has higher correlation across all human metrics more than any other metric proposed to-date. **Notes:** -U: Calculated on user response, -B: Calculated on bot response, -U/B: Calculated between user and bot response, -B/B: Calculated between consecutive bot utterances.

## A.7 Embedding-based metrics

**Embedding Average** Taking the mean word embedding of the generated sentence $e_g$ and the target sentence $e_t$, the embedding average metric is the cosine distance between the two.

$$\bar{e}_t = \frac{\sum_{w \in t} e_w}{|\sum_{w' \in t} e_{w'}|} \tag{1}$$

$$\text{Avg}(\hat{e}_t, \hat{e}_g) = cos(\bar{e}_t, \bar{e}_g) \tag{2}$$

**Vector Extrema** The extrema vector for a sentence can be calculated by taking the most extreme value for each dimension ($e_w^{(d)}$) among the word vectors in the sentence. The extrema embedding metric is again the cosine between the extrema sentence vectors.

$$\hat{e}_t^{(d)} = \begin{cases} \max_{w \in t} e_w^{(d)} & \text{if } e^{(d)} > |\min_{w' \in t} e_{w'}^{(d)}| \\ \min_{w \in t} e_w^{(d)} & \text{otherwise} \end{cases} \tag{3}$$

$$\text{Ext}(\hat{e}_t, \hat{e}_g) = cos(\hat{e}_t, \hat{e}_g) \tag{4}$$

**Greedy Matching** The greedy matching distance is computed by matching word vectors in a source sentence ($s$) with the closest words vectors in the target sentence($s$).

$$G(r, \hat{r}) = \frac{\sum_{w \in r;} \max_{\hat{w} \in \hat{r}} cos(e_w, e_{\hat{w}})}{|r|} \tag{5}$$

$$\text{Grd}(s, t) = \frac{G(s, t) + G(t, s)}{2} \tag{6}$$

## A.8 Static evaluation setup details

We replicated the static evaluation found in previous work [1, 2]. We sampled conversation contexts from the test set of each corpus and generated samples by each model based on these contexts. After filtering by context length (>10 tokens) and removing contexts which contain <unknown>tokens, we sampled 100 examples. We divided each set of 100 examples into two batches of 50 for annotators to rate. Annotators recruited through Amazon Mechanical Turk were first trained with an example question. Annotators must be in the United States and had to correctly answer all training questions before beginning the task. Figure A.6 shows the interface displayed to crowdworkers in the static evaluation task. We asked annotators to select which sentence was better for quality, fluency, relatedness, and empathy. Note that in static single-turn evaluation, annotators only rate a single

Please read the conversation and answer which response is better:

**Context**:
Person 1: when are you leaving?
Person 2: tomorrow.
Person 1: i'm going to miss you.
Person 2: that's what you said the other night.

**Response A** : well, i mean it more now.
**Response B** : i don't know.

|  | Response A is better | Response B is better | About the same |
|---|---|---|---|
| Which response would you rate higher in **quality**? | ○ | ○ | ○ |
| Which response is more **fluent** ? *i.e. better grammar and sentence structure* | ○ | ○ | ○ |
| Which response is more **related** to the conversation context? | ○ | ○ | ○ |
| Which response is more **empathetic** ? *i.e. more supportive of the people speaking in the conversation context* | ○ | ○ | ○ |

Figure A.6: Static single-turn evaluation interface crowdworkers see.

Table A.2: Results from human static evaluation for EI vs. Baseline models for HRED, VHRED, and VHCR models across quality, fluency, relatedness and empathy pairwise comparisons with 90% confidence intervals

| Model | Metric | Cornell | | | Reddit | | |
|---|---|---|---|---|---|---|---|
| | | Wins % | Losses % | Ties % | Wins % | Losses % | Ties % |
| HRED | quality | **40.8** ± 4.9 | 24.5 ± 4.9 | 34.8 ± 9.2 | **31.3** ± 5.2 | 29.5 ± 6.6 | 39.3 ± 10.7 |
| | fluency | 10.3 ± 4.4 | **17.3** ± 4.1 | 72.5 ± 8.1 | **22.8** ± 5.3 | 20.0 ± 7.1 | 57.3 ± 11.2 |
| | relatedness | 36.3 ± 6.5 | 28.7 ± 4.8 | 35.0 ± 7.9 | **34.3** ± 2.8 | 30.3 ± 7.8 | 35.5 ± 9.7 |
| | empathy | **37.8** ± 7.2 | 24.5 ± 5.6 | 37.8 ± 8.9 | **32.5** ± 3.4 | 31.2 ± 5.9 | 36.3 ± 8.0 |
| VHRED | quality | **36.9** ± 4.7 | 36.6 ± 5.6 | 26.6 ± 6.9 | **39.0** ± 7.0 | 34.0 ± 5.3 | 27.0 ± 8.9 |
| | fluency | 23.4 ± 9.6 | **27.7** ± 8.3 | 48.9 ± 16.3 | **29.0** ± 13.6 | 23.3 ± 9.3 | 47.7 ± 21.6 |
| | relatedness | **37.4** ± 5.4 | 33.1 ± 7.2 | 29.7 ± 9.6 | **38.3** ± 5.6 | 33.0 ± 5.1 | 28.7 ± 9.0 |
| | empathy | **36.6** ± 9.4 | 34.0 ± 8.4 | 29.4 ± 15.8 | **34.7** ± 8.7 | 33.7 ± 6.7 | 31.7 ± 10.9 |
| VHCR | quality | **33.0** ± 6.1 | 29.0 ± 5.4 | 38.0 ± 10.1 | **33.7** ± 7.9 | 27.3 ± 3.3 | 39.0 ± 8.6 |
| | fluency | 13.5 ± 4.1 | **25.5** ± 4.3 | 66.0 ± 7.7 | **24.7** ± 7.2 | 18.3 ± 5.2 | 57.0 ± 10.2 |
| | relatedness | **40.8** ± 4.8 | 26.8 ± 6.8 | 32.5 ± 10.5 | 28.3 ± 6.6 | **31.3** ± 3.6 | 40.3 ± 8.4 |
| | empathy | **32.8** ± 6.6 | 28.0 ± 7.8 | 39.3 ± 13.7 | **30.3** ± 3.9 | 24.0 ± 4.6 | 45.7 ± 7.6 |

bot-generated response; thus they cannot judge the diversity of response generation in the dialog model and only rate the remaining four qualities. Table A.2 summarizes the results for all 4 metrics and is an uncondensed version of table 4. One notable exception to the pattern of EI models winning is fluency; baseline models trained on the CORNELL corpus generated more fluency wins.

Noting the high disagreement between annotators in this task, we further examined the ambiguous examples in the human evaluation test set. We define an ambiguous example as a question where an equal number of annotators select the first sentence as better as the second sentence. If the two examples were similar, annotators would select the "tied" option. An equal number of selections for each answer as the winner indicates a disagreement in perception. Table A.3 summarizes the number of ambiguous examples per model and metric out of 100 in total for each box. After removing these ambiguous example from calculating wins, losses and ties, the results are similar to table A.2. The number of ambiguous samples further highlights the noisy and unreliable nature of static single-turn evaluation.

Table A.3: Count of ambiguous examples in human static evaluation.

| | Cornell | | | Reddit | | |
|---|---|---|---|---|---|---|
| | HRED | VHRED | VHCR | HRED | VHRED | VHCR |
| Quality | 12 | 13 | 15 | 26 | 15 | 9 |
| Fluency | 4 | 10 | 10 | 12 | 20 | 6 |
| Relatedness | 11 | 12 | 10 | 15 | 13 | 7 |
| Empathy | 16 | 9 | 12 | 14 | 17 | 7 |

Figure A.7: Interactive evaluation chat interface

## A.9 Interactive evaluation details

For our interactive evaluation, we built a platform to mimic a natural chat setting. Figure A.7 is an example conversation within the platform that interactive evaluation participants see. Annotators can optionally click the up and down arrows beside each chatbot response to give feedback on the specific utterance. Once 3 or more turns of the conversation has taken place, participants may click "Close Chat and Rate". This will take them to the rating page where the conversation to be rated is presented along side the 7 point Likert scale questions used to asses the conversation (Figure 2).

Participants both from Amazon Mechanical Turk and from the authors' institution were recruited for interactive evaluation. Although the minimum required number of turns is 3, the average number of responses per conversation of participants varied between 3.00-10.58 turns with the average at 5.43 turns. Table A.4 summarizes the number of ratings collected for each model.

The average rating each annotator gave differed significantly between annotators. As a result, we also computed scores for interactive evaluation after normalizing each annotator's scores. We restricted ratings down to only annotators who completed 10 or more ratings which left 301 ratings. Similar to Table 2, the mean ratings for EI (Emotion+Infersent) models were higher than the mean ratings for the baseline models.

## A.10 Website server setup and configuration

The server was hosted on a Google Cloud Platform virtual instance with 64GB of RAM and a NVIDIA Tesla P100 graphics card. The backend was a Django program being served by NGINX and uWSGI. For simplicity, we opted to have the Django process import the chatbots into the same Python process as Django, rather than have the two connect to each other via other means such as

Table A.4: Summary table of ratings collected per model.

| | Cornell | | | Reddit | | |
|---|---|---|---|---|---|---|
| | HRED | VHRED | VHCR | HRED | VHRED | VHCR |
| Baseline | 55 | 46 | 53 | 55 | 36 | 39 |
| EI | 49 | 39 | 42 | 56 | 44 | 52 |

Figure A.8: (a) 64-most frequent emojis as predicted by [3] used for calculating emotion embeddings. (b) Assigned weights used for reducing the 64-dimensional emotion embedding into a *Sentiment* score.

sockets. This configuration decreased development time and increased reliability, but it would need to be revisited if the server needed to scale several orders of magnitude past what was required for this study. The current configuration was still able to support hundreds of simultaneous users and host more than 30 bots concurrently.

The chatbots were kept in a separate project from the Django project and maintained separately from the server code. Each chatbot extended an abstract class that defined key methods for the Django program to use, and was registered to a globally accessible dictionary via a decorator. The Django project was provided the path to the Chatbots project in its PYTHONPATH, so it could import the dictionary in which all the chatbot objects had been registered and use that to dynamically determine which chatbots were available and to access them in its views.

It is important to note that the chatbots used PyCUDA, and PyCUDA does not work in a multiprocessing environment. Because of this, uWSGI needed to be configured to only have one python process and to disable any attempt at multiprocessing. Furthermore, the chatbots required substantial startup times, so all chatbots are kept in memory at all times in the Django process. In order to keep all the chatbots in memory concurrently, we needed a very high amount of RAM on our server and opted for a 64GB virtual instance, and a GPU with 16GB RAM. This combination of CUDA to run the chatbots on the GPU with a high amount of RAM to keep all bots in memory at the same time resulted in incredibly fast server response times, with effectively no increase in response time when using the bots in requests compared to requests that did not.

For further information and instructions on server configuration, please read the server documentation available at `https://github.com/asmadotgh/neural_chat_web`.

### A.11 Emotion embedding details

We calculate emotion embeddings of an utterance using a using a state-of-the-art sentiment-detection model [3]. This pre-trained model outputs a probability distribution over 64 most-frequently used emojis as presented in [3]). We define a set of weights over the emojis and calculate the weighted sum over an emotion embedding vector to derive a *Sentiment* score which is higher for positive sentiment and lower for negative sentiment (See Figure A.8).

### A.12 Hyper-parameter tuning details

For the baseline models that were trained on the CORNELL dataset, we used the parameters reported in [4, 1, 2] that achieved state-of-the-art results for HRED, VHRED, and VHCR models trained on the same dataset, respectively. For EI models, we compared a combination of values for encoder hidden size (400, 600, 800, 1250), decoder hidden size (400, 600, 800, 1250), context size (1000, 1250), embedding size (300, 400, 500), word drop (0, .25), sentence drop (0, .25), beam size (1, 5). Learning rate (.0001), dropout (.2) were fixed. Batch size 80 was used. If due to memory limitation the job was not successfully completed, batch size 64 was used. Additionally, we tuned the EI parameters, i.e., emotion weight (25, 150), infersent weight (25K, 30K, 50K, 100K), emotion sizes (64, 128, 256), infersent sizes (128, 1000, 2000, 4000). Due to limited computational resources, we were not able to

Table A.5: Hyper-parameters used for different models.

| Dataset | Version | Model | Batch size | Dropout | Decoder hidden size | Encoder hidden size | Context size | Embedding size | Word drop | Sentence drop | Beam size | Emotion weight | Emotion discriminator layer size | Infersent weight | Infersent discriminator layer size |
|---|---|---|---|---|---|---|---|---|---|---|---|---|---|---|---|
| Cornell | Baseline | HRED | 80 | .2 | 400 | 400 | 1000 | 300 | .0 | .0 | 5 | - | - | - | - |
| | | VHRED | 80 | .0 | 1000 | 1000 | 1000 | 400 | .25 | .0 | 5 | - | - | - | - |
| | | VHCR | 80 | .2 | 1000 | 1000 | 1000 | 500 | .25 | .25 | 5 | - | - | - | - |
| | EI | HRED | 64 | .2 | 1000 | 1000 | 1000 | 500 | .0 | .0 | 1 | 25 | 128 | 100K | 4000 |
| | | VHRED | 80 | .2 | 1250 | 1250 | 1000 | 600 | .0 | .0 | 1 | 25 | 128 | 30K | 128 |
| | | VHCR | 32 | .2 | 1000 | 1000 | 1250 | 600 | .0 | .0 | 1 | 25 | 128 | 25K | 4000 |
| Reddit | Baseline | HRED | 64 | .2 | 1000 | 1000 | 1000 | 500 | .0 | .0 | 1 | - | - | - | - |
| | | VHRED | 32 | .2 | 1250 | 1250 | 1000 | 600 | .0 | .0 | 1 | - | - | - | - |
| | | VHCR | 32 | .2 | 1000 | 1000 | 1250 | 600 | .0 | .25 | 1 | - | - | - | - |
| | EI | HRED | 64 | .2 | 1000 | 1000 | 1000 | 500 | .0 | .0 | 1 | 25 | 128 | 25K | 2000 |
| | | VHRED | 32 | .2 | 1250 | 1250 | 1250 | 600 | .0 | .0 | 1 | 25 | 128 | 100K | 4000 |
| | | VHCR | 32 | .2 | 1000 | 1000 | 1250 | 600 | .0 | .0 | 1 | 25 | 128 | 100K | 4000 |

run a grid search on the aforementioned values. Instead we used combinations of the parameters that heuristically were more viable.

For the models that were trained on the REDDIT dataset, a set of properly tuned baseline parameters were non-existent. Thus, to ensure fair comparison, we used a similar approach for baseline and EI hyper-parameter tuning: We explored a combination of values for encoder hidden size (400, 600, 800, 1250), decoder hidden size (400, 600, 800, 1250), context size (1000, 1250), embedding size (300, 400, 500, 600), word drop (0, .25), sentence drop (0, .1, .25), and beam size (1, 5). Learning rate (.0001), dropout (.2) were fixed. Batch size 64 was used. If due to memory limitation the job was not successfully completed, batch size 32 was used. Due to limited computational resources, we were not able to run a grid search on all the aforementioned values. Instead we used combinations of the parameters that heuristically were more viable. To ensure fair comparison, any selected combination was tested for both baseline and EI models. Then, for EI models, we tuned the parameters that were solely relevant to the EI design, such as the weight of emotion and infersent term in the loss function and the size of the added discriminator networks: Emotion weight (25), infersent weight (25K, 50K, 100K), emotion sizes (64, 128, 256), infersent sizes (100, 128, 1000, 2000, 4000). See Table A.5 for a summary of the final selected parameters.

## A.13 Self-Play Overlap Analysis

| Model | Version | Cornell | | Reddit | |
|---|---|---|---|---|---|
| | | 3-turn overlap | 5-turn overlap | 3-turn overlap | 5-turn overlap |
| HRED | baseline | 19.49% | 1.76% | 2.02% | 0.24% |
| | EI | 6.48% | 0.30% | 2.12% | 0.16% |
| VHRED | baseline | 0% | 0% | 0% | 0% |
| | EI | 0.16% | 0% | 0.16% | 0% |
| VHCR | baseline | 0% | 0% | 0% | 0% |
| | EI | 0% | 0% | 0% | 0% |

Table A.6: Percentage of pairs of conversations in each 100 sample for each model where there are 3 or 5 consecutive conversation turns that are exactly the same.

As a post-hoc sanity check on the conversations generated from self-play, we check whether there is i) overlap among generated conversations, and ii) overlap between these conversations and the training set. High overlap among generated conversations would indicate that there is a lack of diversity in

| | | Cornell | | Reddit | |
|---|---|---|---|---|---|
| Model | Version | 2-turn overlap | 3-turn overlap | 2-turn overlap | 3-turn overlap |
| HRED | baseline | 58% | 0% | 0% | 0% |
| | EI | 65% | 0% | 0% | 0% |
| VHRED | baseline | 8% | 0% | 5% | 0% |
| | EI | 5% | 0% | 12% | 0% |
| VHCR | baseline | 4% | 0% | 4% | 0% |
| | EI | 3% | 0% | 3% | 0% |

Table A.7: Percentage of of conversations (100 sample for each model) where there are 2 or 3 consecutive conversation turns that match the training set.

the conversations generated by self-play while high overlap with the training set suggests self-play may be memorizing training dialog.

To measure overlap between the 100 conversations generated in each model, we consider all 3 and 5 consecutive conversational turns over the 10 turns in each conversation. We compare each pair of conversations in the 100 generated conversations in total to compute a percentage of conversations which contain overlap in this pairwise comparison. Table A.6 summarizes these results and illustrates that overlap is not significant for most models. The exception is the non-variational models trained on the Cornell corpus (e.g. HRED Cornell). Qualitative evaluation reveals that these are degenerate cases where "what?" or "I don't know" or "I'm sorry" are repeated multiple turns.

To measure repetition with respect to the training set, we take all 2-turn and 3-turn windows in the self-play generated conversations and compare with the entire training set to check whether there is overlap. Table A.7 shows the percentage of conversations (100 total for each model) where there is a 2-turn or 3-turn dialog appearing exactly in the training set. Since each conversation is 10 turns long, all of the conversations are distinct from the training set and no conversation contains more than 2-turns of overlap with the training set. The 2-turn overlap again appears due to cases where "what?" and "hi" are repeated for 2 turns.