[Reviews · NeurIPS 2019]

Reviewer 1



Originality/significance/quality: The paper addresses an important topic, as there is indeed a need for better evaluation of dialog systems. The paper attempts to move away from traditional evaluation of open-domain dialog systems (i.e., judge response given its conversation history) and moves towards a more interactive one (i.e., human talking to a bot), which is likely an important step towards better evaluation. However, I do have several serious concerns about this work in its current form: (1) The authors contrast their work with existing evaluation for open-domain dialog evaluation, which they call “single-turn” evaluation. They point out that this type of evaluation prevents it from capturing “failure modes […] such as a lack of diversity in the responses, inability to track long-term aspects of the conversation”. I think this is rather misleading and the term is “single-turn” is a misnomer. Most previous work has indeed evaluated each conversation by factorizing it into a sequence of independent turn-level judgments, but each of these judgments assesses the quality of the current turn T_n **given** a history of several previous turns …, T_n-k, … T_n-1. As opposed to what “one-turn” implies, judges look at several conversational turns at once for each judgment. While one could set n-k=1 and therefore let the judge see the entire conversation T_1,…T_n (and therefore let the judge identify “lack of diversity”, etc.), the practice is to set k to a smaller window as to reduce cognitive overload for the judge and therefore make evaluation more reliable. Much of the earlier work on open-domain dialog system used a small context window (k=2 or 3, which should really be called “two-“ or “three-turn evaluation”), but more recent work (e.g., Serban et al.) has used much bigger windows of context and values of k. Serban et al. and others would not have been able to show that longer context and better representation thereof helped without the use of *multi-turn* evaluation. In sum, I think the characterization of previous work in the paper is rather misleading and I think most of what is said about “single-turn” in the paper would have to be rewritten. (2) Self-play protocol: It seems to me that this automatic evaluation is easily gameable, as each prior turn T_n-1 is generated by the same system, so the system could “strategically” only ask questions or issue queries for which it has high-quality responses. For example, one could design a system that ignores the original turn T_1 and then (for T_2, …, T_n) simply regurgitates human-to-human conversations memorized from the training data. While one might argue that gamebility is only relevant in the context of a competition and that one could decide to leave that issue to future work, I think it is important to make metrics resistant to gaming as it is fairly common to optimize directly or indirectly for automatic metrics (e.g., directly through RL as in https://arxiv.org/abs/1511.06732). Even if most system designers do not deliberately try to game metrics, RL or repeated queries of the metric might just take care of that, so metrics need to be gaming-proof from the outset. (3) I am not convinced by the comparisons in the paper between “single-turn” evaluation and the self-play scenario of the paper. Results are not comparable, as the paper uses Kappa for single turn and Pearson’s r for self-play. As in previous work, it is totally possible to use Pearson’s r for assessing the level of correlation between existing metrics and human judgment, so why haven’t the authors done that? (In a related note, Spearman’s rho or Kendall’s tau are usually deemed more robust than Pearson’s r as they don’t assume that correlation is necessarily linear). (4) Semantic and engagement metrics: this is more a minor concern, but these metrics are potentially gameable too as they are machine-learned. We can think of it this way: trainable metrics are akin to one turn of GAN training, where the generator is the response generation system and the discriminator is the learned metric. Clearly, one would have to iterate multiple times (GAN style) as otherwise one could modify the generator to find the weak spots in which the discriminator can easily be fooled. This intuition of GAN for dialog evaluation is described in more details in (e.g.) https://arxiv.org/abs/1701.08198, https://arxiv.org/pdf/1701.06547.pdf, https://arxiv.org/pdf/1809.08267.pdf. Clarity: the paper is mostly clear.

Reviewer 2



This paper addresses the problem of evaluating spoken dialogue systems, pointing out that current methods which focus on evaluating a single turn given the history bear little relation to the overall user experience. The core idea is a hybrid metric M_h which is a linear combination of three metric classes, each being measured across a whole dialogue: sentiment using a classifier trained on Twitter, semantic similarity measured by Infersent plus a number of simple word based metrics, and Engagement based a question score and length of user inputs. The dialogues themselves are generated automatically by self-play ie the systems output is simply fed back as input for a fixed number of turns. The ideas underlying M_h are then used in two ways. Firstly its main components: sentiment and semantics are used to regularise training a standard HRED. VHRED and VHCR dialogue model. This is shown to improve human evaluations of quality, fluency, diversity, contingency and empathy in most cases. They then show that although single turn metrics show similar trends they are very noisy and correlation between with human evaluation is poor. However, the proposed M_h metric correlates very well with human judgement. Overall, this is an interesting paper. The proposed metric appears to be very useful and the self-play idea is useful where there is no existing dialogue data to evaluate on (as would be the case during system development). However, this is really two papers in one. Although they are inspired by the same idea, the proposed regularisation for training chatbots is really a separate topic to evaluation and I would have preferred the focus to be on the latter. In particular, it isnt clear why the proposed metric could not be applied to some of the existing published data sets eg Multi-Woz.

Reviewer 3



The paper starts out with an attempt to address the difficulty of evaluating a conversational model that is supposed to be able to produce an interesting dialog (with human) in an open-domain chat. This is indeed a challenge for the research on automated dialog systems. However the paper soon goes into the addition of an EI regularization layer, which is confusing and somewhat distracts from the main focus on the evaluation system. In the experiments it is shown that using five conventional metrics with humans doing the scoring, the proposed EI regularization can improve almost each of the 3 known models; yet between the 3 models the ANOVA results do not show significant differences. Then using the hybrid metric and self-play automation, it is shown that the addition of the EI layer is favorable over the baseline models, and the results correlate well with human judgment using the conventional metrics on interactive sessions. One critical discussion that is missing is in what specific ways self-play dialog differs from interactions with a real user. Does self-play assume that the same persona is talking on both sides? How can one accurately represent the persistent but potentially very different personas represented by real users? The paper also seems to struggle between two main proposals: one on the addition of the EI regularization layer, and the other on advocating self-play automated multi-turn evaluation and a hybrid metric. If the EI proposal is meant to be the main contribution, the arguments about the evaluation metrics and self-play set up are problematic, because you are showing that with your proposed evaluation, your proposed modification is better than the baseline. There is a question on the fairness of the evaluation choices. On the other hand, if the evaluation system is meant to be the main proposal, then the choice of the EI regularization could be an unfair demonstration of the effectiveness of the evaluation system. How are you sure that your proposed evaluation scheme will not contradict human judgment when a different set of alternatives are evaluated? Either proposal provides some useful but rather incremental contribution to the state of the art in dialog modeling. Given that, the paper seems to be a better fit for an audience more specialized in dialog modeling. It is unclear how the broader audience in NeurIPS can be benefited from the work.

[Author Response · NeurIPS 2019]

We thank the reviewers for their time and detailed reviews. We address the comments by clarifying misunderstandings and providing further evidence of this work's significance.

**1. Multi-turn evaluation [R1]:** We believe there has been a major misunderstanding. We acknowledge that (Serban et al., 2016) and (Park et al., 2018) use multiple turns of *context* $T_{n-k}, \ldots, T_{n-1}$, and "generate the next [1 or 3] consecutive utterances". We use multiple turns of context (i.e. $T_1, \ldots, T_{n-1}$) to generate a single bot response and name this *single-turn*. To clarify, we will name this *static* evaluation in our paper instead. Our discussion of previous work remains valid after this clarification and substitution of terms. We use *multi-turn* to refer to *interactive* evaluation, where the dynamic context comes from humans input (i.e. this is different from generating three consecutive utterances). After multiple alternating real-time human input and bot-generated turns, we ask annotators to make a holistic evaluation of their conversational chat experience; this is the test of generalization we propose. Our current nomenclature was motivated by the necessity of multiple interactive conversation turns. We thank R1 for pointing out this potential for misunderstanding and will use *interactive* evaluation moving forward.

**2. Gameability [R1]:** The purpose of the self-play metric is post-hoc evaluation of a dialog model, rather than to be optimized for while training. Reward exploitation in RL is a known problem and an active area of research (Amodei et al., 2016). One of the methods to alleviate that is using multiple

Table 1: Interactive human evaluation of different reward functions with RL

| Reward | Quality | Fluency | Diversity | Contingen. | Empathy | Total |
|---|---|---|---|---|---|---|
| Conv. len. | 2.20 ±.40 | 3.61 ±.53 | 3.02 ±.52 | 2.25 ±.46 | 2.48 ±.45 | 13.57 ±1.84 |
| Semantic sim. | 1.93 ±.34 | 3.50 ±.45 | 2.37 ±.45 | 2.11 ±.45 | 2.52 ±.48 | 12.43 ±1.75 |
| Laughter | 1.96 ±.38 | 3.56 ±.48 | 2.33 ±.51 | 1.93 ±.42 | 3.20 ±.55 | 12.98 ±1.60 |
| # Words | 2.11 ±.32 | 3.96 ±.44 | 3.04 ±.45 | 2.04 ±.35 | 2.55 ±.46 | 13.70 ±1.44 |
| Sent. trans. | 2.02 ±.31 | 3.71 ±.49 | 2.98 ±.50 | 2.04 ±.42 | 2.84 ±.48 | 13.60 ±1.63 |
| Question | 2.29 ±.37 | **4.31 ±.50** | 3.31 ±.52 | 2.20 ±.40 | 2.60 ±.41 | 14.71 ±1.63 |
| Sentiment | **2.47 ±.32** | 4.05 ±.45 | 3.23 ±.46 | **2.42 ±.39** | **3.23 ±.55** | **15.40 ±1.49** |
| VHCR-cornell | 2.13±.25 | 2.68±.31 | **3.75±0.35** | 2.19±.27 | 2.34±.32 | 13.09±1.02 |

rewards with conflicting objectives (Kalyanmoy, 2014). $M_H$ is a hybrid of conflicting objectives and thus is less susceptible to exploitation, as shown by the learned $\lambda$s in Figure 1 in supplementary materials. Additionally, we have run further experiments and provide strong empirical evidence that our proposed metrics are not easily exploitable. As shown in Table 1, we have successfully used these rewards to learn with a batch RL Q-learning (Fujimoto et al., 2018) improved with KL-control to penalize divergence from a pre-trained language model (Abdolmaleki et al., 2018). Interactive human-evaluation reveals that many of these models outperform VHCR-Cornell baseline (Park et al. 2018) in several aspects. However, we acknowledge that our current work does not *prove* that these metrics are robust to adversaries. This is an open research area (e.g. it has not yet been demonstrated how convex bounds can be used on text representations (Wong and Kolter, 2018)), and out of scope for this paper. We will extend the discussion to highlight caveats and precautions for when our evaluation framework is used beyond its intended purpose.

**3. Primary (evaluation) and secondary (EI) contributions [R2, R3]:** The main contribution of this work is an evaluation methodology that captures higher level human conversation concepts. The reasons why we included EI models in the same paper are: 1) EI models are intended to promote awareness to higher level human conversation concepts; 2) EI results in significantly different models based on human judgment; 3) To showcase the effectiveness of an evaluation methodology, a pool of models with significantly different qualities are needed. We have made sure that there are no circular arguments in our evaluation: 1) we use traditional *static* evaluation that shows improvements using EI regularization; 2) the main criteria in evaluation, showing significant differences, is *interactive* human judgments of quality. Humans are blind to the model, EI, or dataset type; 3) our self-play evaluation methodology captures human judgment afterwards, rather than being used as the primary evidence for enhanced quality in EI models. We will revise the introduction to emphasize the main contribution and clarify the reasons EI models are included.

**4. Platform [R1]:** Releasing our code and platform is a side contribution for transparency and reproducibility. Also, it will add diversity to the platforms future practitioners can choose to use. Following R1 comments, we will reference other platforms in the related work; however, their thorough review is beyond the scope of this paper.

**5. Correlation metrics [R1]:** To clarify a potential misunderstanding: we followed (Park et al., 2018) and (Serban et al., 2016) to use categorical wins/losses in traditional *static* (new nomenclature for the single-turn evaluation, see item 1) evaluation rather than Likert scale; Cohen's $\kappa$ has been used to compare *inter-rater agreement* across MTurkers and to contrast our observation with previous work, e.g. (Lowe et al., 2018). We use *static* evaluation to benchmark EI models against

Table 2: self-play vs interactive eval.

| Interactive Eval. | Spearman ($\rho, p$) | Kendall ($\tau, p$) |
|---|---|---|
| Quality | (0.68, 0.02) | (0.45, 0.04) |
| Fluency | (0.42, 0.17) | (0.18, 0.46) |
| Diversity | (0.64, 0.03) | (0.48, 0.03) |
| Contingency | (0.16, 0.62) | (0.12, 0.64) |
| Empathy | (0.76, 0.00) | (0.55, 0.01) |

HRED/VHRED/VHCR to motivate adding EI models to the pool of models we compare in *interactive* evaluation. Motivated by the potential failure modes of *static* evaluation, we propose *interactive* evaluation. We introduce self-play, then compare existing metrics to human judgments on *interactive* evaluation in Figure 5 in the paper. We will add additional statistics (Table 2) that further strengthen our findings.

**6. Future Work [R2, R3]:** R2: Adapting our open-domain evaluation to goal-oriented bots is an interesting direction towards measuring dialog experience (e.g. empathy) beyond accomplishing the primary task. R3: Extending self-play to multiple personas is an interesting next step that can be achieved through training on, for example, different sub-reddits (Mazare et al., 2018). We will include discussion on how this can be incorporated in future self-play settings.

[Meta-Review · NeurIPS 2019]

This paper explores interesting directions, in particular 1) using interactive settings to evaluate a model rather than a single answer, and 2) combining different automated metrics in a weighted sums to approximate human evaluation (e.g., based on sentiment). Reviewers have raised crucial points, regarding gameability (so that using the metrics for training a model is tricky if not followed by a non-gameable evaluation), and lack of comparability between different self-play. It’s indeed a much better evaluation setting if the system does not control both sides (e.g., models being matched to the same set of fixed models), so authors should definitely follow that direction. However, I expect this work would still be interesting to the dialog community: many of the diagnostic advantages of the model-talking-to-model setting remain, in practice, especially because the model is in fact not trained with the self-play objective, but that criterion is only used post hoc (so the system can’t extensively exploit it during training). In practice, a lot of the problems of the generations of a given model already show up during self-play, and the reasonable worry raised by reviewers that the model could exploit the metric remains theoretical at the moment. So, with the caveat that the results in self-play mode may be too optimistic, it can still act as a useful and cheaper (compared to humans responding) diagnostic tool to catch some issues like excessive repetitions, etc. Authors should make sure to at the very least include discussion of the crucial caveats raised by the reviewers (e.g. that a more reliable evaluation would be evaluating a model paired to a stable "opponent/user model"), or better, results in that set-up.